# Portable Wideband Directional Antenna Scheme with Semicircular Corrugated Reflector for Digital Television Reception

**DOI:** 10.3390/s22145338

**Published:** 2022-07-17

**Authors:** Bancha Luadang, Rerkchai Pukraksa, Pisit Janpangngern, Khanet Pookkapund, Sitthichai Dentri, Sompol Kosulvit, Chuwong Phongcharoenpanich

**Affiliations:** 1Faculty of Engineering, Rajamangala University of Technology Rattanakosin, Nakhon Pathom 73170, Thailand; bancha.lua@rmutr.ac.th; 2School of Engineering, King Mongkut’s Institute of Technology Ladkrabang, Bangkok 10520, Thailand; 62601099@kmitl.ac.th (R.P.); pisit.janpangngern@gmail.com (P.J.); sompol.antenna@gmail.com (S.K.); 3Faculty of Science and Technology, Phranakhon Si Ayutthaya Rajabhat University, Phranakhon Si Ayutthaya 13000, Thailand; pkhanet@aru.ac.th; 4College of Industrial Technology, King Mongkut’s University of Technology North Bangkok, Bangkok 10800, Thailand; sitthichaid@kmutnb.ac.th

**Keywords:** corrugated reflector, digital television, portable antenna, radome, wideband antenna

## Abstract

This research proposed a portable wideband horizontally-polarized directional antenna scheme with a radome for digital terrestrial television reception. The operating frequency band of the proposed antenna scheme is 470–890 MHz. The portable antenna scheme was an adaptation of the Yagi-Uda antenna, consisting of a folded bowtie radiator, a semicircular corrugated reflector, and a V-shaped director. Simulations were carried out, and an antenna prototype was fabricated. To validate, experiments were undertaken to assess the antenna performance, including the impedance bandwidth (|*S*_11_| ≤ −10 dB), gain, and unidirectionality. The measured impedance bandwidth was 75.93%, covering 424–943 MHz, with a measured antenna gain of 2.69–4.84 dBi. The radiation pattern was of unidirectionality for the entire operating frequency band. The measured xz- and yz-plane half-power beamwidths were 150°, 159°, 160° and 102°, 78°, 102° at 470, 680, and 890 MHz, with the corresponding cross-polarization below −20 dB and −40 dB. The radome had a negligible impact on the impedance bandwidth, gain, and radiation pattern. The power obtained for the outdoor test, at 514 MHz, was 38.4 dBµV (−70.4 dBm) with a carrier-to-noise ratio (C/N) of 11.6 dB. In addition, the power obtained for the indoor test was 26.6 dBµV (−82.2 dBm) with a C/N of 10.9 dB. The novelty of this research lies in the concurrent use of the Yagi-Uda and bowtie antenna technologies to improve the impedance bandwidth and directionality of the antenna for digital terrestrial television reception.

## 1. Introduction

In recent years, attempts have been made to synthesize diverse antenna technologies to improve the impedance bandwidth of antennas for digital television reception. Specifically, in [1], a wideband dipole antenna for digital television reception was proposed using a multi-loop radiator and a coplanar waveguide feed. In [2], an antenna with seven resonance pads on the ground plane was proposed to improve the impedance bandwidth.

In [3], an internal antenna that combined a planar inverted F antenna (PIFA) and a loop antenna for digital television reception achieved an impedance bandwidth of 48%, covering 470–771 MHz. In [4], a miniaturized wideband meander-line antenna using a magneto dielectric material was developed for the DVB-H/LTE13/GSM850/900 applications. Despite wide impedance bandwidth [1,2,3,4], these antennas suffer from low gain.

As a result, further attempts were made to develop antennas for digital television reception with wide impedance bandwidth, high gain, and stable radiation pattern throughout the operating frequency band (470–890 MHz). Specifically, the bowtie antenna technology was adopted in the development of antennas for digital television reception due to impedance bandwidth enhancement [5]. In addition, the Yagi-Uda antenna technology was deployed to achieve high gain and directionality [6].

For the bowtie antenna technology, a U-shaped bowtie magneto-electric dipole unidirectional antenna with a dual-layered horn reflector was proposed for ultra-wideband applications [7]. In [8], multiple bowtie patch antenna configurations with broadband operation, adjustable beamwidth, and multiple-beam capability were proposed for broadband indoor wireless communication. In [9], an adaptive ground penetrating radar (GPR) antenna based on bowtie antenna technology was proposed for the classification of soil types.

In [10], an antenna with a bowtie dipole element above the ground plane and a metallic bridge for wide beamwidth was proposed for base station applications. A tapered bowtie slot antenna [11] and a cavity-backed bowtie antenna [12] achieved the unidirectional radiation pattern. In [13], a substrate integrated waveguide (SIW) cavity backed bowtie slot antenna was proposed to improve the gain and unidirectionality. In addition, a unidirectional bowtie array antenna with an incision gap for digital video broadcasting-T2 base station applications was proposed in [14].

The Yagi-Uda antenna is conventionally used for television signal reception, and it is normally installed on the rooftop. The advantages of the Yagi-Uda antenna are its simple design, low cost, and reasonably high gain. The principal elements of the Yagi-Uda antenna include the radiator, the reflector, and the director [15]. In [16,17], the broadband Yagi-Uda antennas were proposed for wireless communication applications.

Currently, the antennas commonly used for digital television signal reception include loop antennas, omnidirectional dipole antennas, monopoles, log-periodic dipole arrays (LPDA), and quasi-Yagi antennas [18]. The loop antennas, omnidirectional dipole antennas, and monopoles can receive signals from every direction but achieve a low gain [19,20,21,22,23]. Meanwhile, the LPDA and quasi-Yagi antennas realize a high gain by using multiple dipoles. However, the unidirectionality of the LPDA and quasi-Yagi antennas gives rise to several installation challenges [24,25,26]. To overcome such shortcomings, this research thus combined the Yagi-Uda and bowtie antenna technologies in the development of the portable wideband directional antenna scheme with a radome for digital television reception.

By definition, a radome is a structural enclosure that protects an antenna and is constructed of materials that minimally attenuate the electromagnetic signal transmitted and received by the antenna [27,28]. The materials used in the radome construction include fiber reinforced plastics (FRP), glass reinforced plastics (GRP), polypropylene (PP), acrylonitrile butadiene styrene (ABS), polystyrene unplasticized polyvinyl chloride (UPVC), and polycarbonate [29,30].

Specifically, this research proposed a portable wideband directional antenna scheme with a radome for digital terrestrial television reception (470–890 MHz). The portable antenna scheme was an adaptation of the Yagi-Uda antenna, consisting of a folded bowtie radiator, a semicircular corrugated reflector, and a V-shaped director. The radome was of ABS material. In the study, simulations were carried out, and an antenna prototype was fabricated. In addition, experiments were undertaken to determine the antenna performance, including the impedance bandwidth (|*S*_11_| ≤ −10 dB), gain, and unidirectionality. The outdoor and indoor testing was also carried out to assess the outdoor and indoor reception performance of the proposed antenna scheme.

The organization of this research is as follows: Section 1 is the introduction. Section 2 details the antenna structure, and Section 3 deals with the parametric study. Section 4 is concerned with the fabrication of the prototype antenna and experimental results. Section 5 discusses the indoor and outdoor application testing. The concluding remarks are provided in Section 6.

## 2. Antenna Structure

Figure 1 shows the geometry of the portable wideband directional antenna scheme for digital terrestrial television reception, and Figure 2 illustrates the proposed antenna scheme of 75 Ω input impedance with radome. The proposed antenna scheme was an adaptation of the Yagi-Uda antenna, consisting of the folded bowtie radiator, semicircular corrugated reflector, and V-shaped director.

The folded bowtie radiator and the corrugated reflector were made of a thin stainless-steel sheet (0.2 mm in thickness). The semicircular reflector was corrugated to reduce the overall physical size of the antenna while increasing its electrical size. In addition, the V-shaped director was made of a hollow aluminum tube 8 mm in diameter. The director was incorporated to improve the directivity and gain of the antenna.

The three elements (i.e., the folded bowtie radiator, corrugated reflector, and V-shaped director) were mounted on a circular plastic supporting plate 165 mm in diameter. The antenna was fed by a coaxial cable connected to a 75 Ω F-type connector, and a balun (λ/4 coaxial balun) was utilized to convert unbalanced to balanced output signal. The portable wideband directional antenna was also covered with a 0.5 mm-thick cylindrical acrylonitrile butadiene styrene (ABS) radome 175 and 209 mm in diameter and height.

Simulations were carried out to optimize the antenna parameters using CST Microwave Studio Suite [31]. Table 1 tabulates the optimal physical dimensions of the portable wideband directional antenna scheme for the 470–890 MHz frequency band. (Note: The experimental results revealed that the ABS radome had a negligible effect on the antenna performance, including the impedance bandwidth, gain, and radiation pattern).

## 3. Parametric Study

### 3.1. Conceptualization of the Portable Directional Antenna Scheme

Figure 3a–c illustrates the conceptualization of the proposed portable wideband directional antenna scheme for digital terrestrial television reception, starting with the Yagi-Uda antenna (model I), the modified antenna with the U-shaped reflector and director (model II), and the proposed antenna scheme (model III).

Model I is the Yagi-Uda antenna of a wire-type structure. The main advantages of the Yagi-Uda antenna are unidirectionality and high gain. However, model I suffered from a narrow impedance bandwidth. Given the center frequency of 680 MHz, the length of the radiator element was λ_C_/2 (220.58 mm), where λ_C_ was the wavelength of the center frequency.

In model II, to improve the impedance bandwidth, the radiator element of model I was replaced with the folded bowtie radiator. The wire reflector was replaced with the stainless-steel U-shaped reflector, and the director was reshaped into the U shape. In model III, the U-shaped reflector was corrugated to reduce the overall physical size of the antenna while increasing the electrical size. Moreover, the corrugated reflector improved the electromagnetic reflection of the proposed antenna scheme [32,33]. The U-shaped director was reshaped into the V shape for the antenna compactness without compromising the antenna performance.

Figure 4 compares the simulated impedance bandwidth (|*S*_11_| ≤ −10 dB) of the antenna models I, II, and III. Model I achieved a narrow impedance bandwidth, covering 620–700 MHz. The narrow impedance bandwidth rendered model I operationally unsuitable for digital television reception. Model II experienced an impedance mismatch between 590–780 MHz, rendering it operationally inapplicable. Model III achieved the simulated impedance bandwidth of 63.85%, covering 468–907 MHz, which encompassed the target frequency band (470–890 MHz). Figure 5 compares the antenna gain of models I, II, and III.

Figure 6 compares the simulated xz- and yz-plane radiation patterns of the antenna models I, II, and III. The three antenna models exhibited similar radiation patterns. However, models I and II failed to cover the target frequency band (Figure 4). The xz- and yz-plane cross-polarizations of model III were below −40 dB and −20 dB, with the corresponding half-power beam width (HPBW) of 149°, 194°, and 158° and 99°, 92.8°, and 84.2° at 470, 680, and 890 MHz, respectively. The xz- and yz-plane back lobe levels were −10 dB.

### 3.2. Evolution of the Portable Wideband Antenna Scheme

This section discusses the evolutionary stages of the proposed portable wideband directional antenna scheme (model III), consisting of three antenna generations: first, second, and third generations. As shown in Figure 7, the first generation was the antenna scheme without a reflector and director, and the second generation was the antenna scheme with the corrugated reflector but without the director. The third generation was the proposed portable wideband directional antenna scheme with the corrugated reflector and the V-shaped director.

Figure 8 compares the simulated impedance bandwidth (|*S*_11_| ≤ −10 dB) of the first-, second-, and third-generation antenna schemes. The first-generation antenna scheme achieved a narrow impedance bandwidth, covering 830–925 MHz, rendering the scheme operationally unsuitable for digital terrestrial television reception. The impedance bandwidth of the second-generation antenna scheme closely resembled that of the third-generation scheme, covering a 468–907 MHz frequency band. Both the second- and third-generation antenna schemes covered the entire target operating frequency band of 470–890 MHz. However, the overall gain of the third-generation antenna scheme was higher than that of the second-generation scheme, as shown in Figure 9.

Figure 10 compares the simulated xz- and yz-plane radiation patterns of the first-, second-, and third-generation antenna schemes. The radiation pattern of the first-generation antenna scheme was of omnidirectionality due to the absence of the corrugated reflector (Figure 7a). The first-generation scheme failed to cover the entire target operating frequency band (Figure 8). The radiation patterns of the second- and third-generation antenna schemes were closely similar. However, the gain of the third-generation scheme was higher than that of the second-generation scheme.

### 3.3. Surface Current Distribution

This section focuses on the third-generation antenna scheme (i.e., the proposed portable wideband directional antenna scheme) due to its wide impedance bandwidth (Figure 8), high gain (Figure 9), and unidirectional radiation pattern (Figure 10).

Figure 11a–c illustrates the simulated surface current distribution of the proposed portable wideband directional antenna scheme at 470, 680, and 890 MHz, respectively. At 470 MHz (the first resonant frequency), the currents are concentrated around the feeding point and along the corrugated reflector, as shown in Figure 11a. In Figure 11b, at 680 MHz (the second resonant frequency), the currents are concentrated around the feeding point, the lower part of the balun, and the upper edge of the corrugated reflector. At 890 MHz (the third resonant frequency), the currents are concentrated around the feeding point, the folded bowtie radiator, and the balun, as shown in Figure 11c. Essentially, the third-generation antenna scheme (i.e., the proposed portable wideband directional antenna scheme) covered the target frequency band for the digital terrestrial television reception of 470–890 MHz, as indicated in Figure 8.

### 3.4. Parametric Sweep of the Antenna Scheme

This section investigates the effect of variable antenna parameters on the impedance bandwidth of the proposed portable wideband directional antenna scheme. The results are graphically presented in Figure 12, Figure 13, Figure 14 and Figure 15.

Figure 12a,b shows the simulated impedance bandwidth (|*S*_11_| ≤ −10 dB) under various widths of the folded bowtie radiator (*W*_di_; 20, 25, 30, 35, and 40 mm) and angles of the folded bowtie radiator (*AG*_dp_; 0, 7.5, 15, 22.5, and 30°). As shown in Figure 12a,b, the optimal *W*_di_ and *AG*_dp_ were 30 mm and 15°, respectively.

In addition, the optimal length of the triangular section of the folded bowtie radiator (*D*_dp_) was 18.7 mm, and the optimal height of the folded bowtie radiator (*h*_mi_) was 15 mm. The optimal width of the first and second sections of the folded bowtie radiator (*W*_mi_) were 15.2 and 15.2 mm, while the optimal width of the final section of the folded bowtie radiator (*W*_r_) was 6.9 mm. The optimal length of one arm of the folded bowtie radiator was thus 101 mm. In other words, the optimal overall length of the folded bowtie radiator was 202 mm.

Figure 13a,b shows the simulated impedance bandwidth (|*S*_11_| ≤ −10 dB) under variable distance from the center of the supporting plate to the reflector (*D*_ref_) and reflector length (*L*_ref_). As shown in Figure 13a,b, the optimal *D*_ref_ and *L*_ref_ were 78.61 mm and 226.42 mm, respectively, covering the target operating frequency band of 470–890 MHz. Figure 13c shows the simulated impedance bandwidth under variable reflector height (*h*_ref_) (25, 45, 65, and 85 mm), and Figure 13d illustrates the corresponding simulated radiation pattern at 890 MHz. The larger *h*_ref_ improved the back lobe level (i.e., front-to-back ratio), especially in the upper frequency band. In Figure 13d, despite the high back lobe level of −21.3 dB under *h*_ref_ = 85 mm, the final antenna was bulky. As a result, the reflector height (*h*_ref_) of 45 mm, with a back lobe level of −14.14 dB, was selected for compactness.

Figure 14a–c shows the simulated impedance bandwidth (|*S*_11_| ≤ −10 dB) under variable distance between the director and the center of the supporting plate (*D*_di_), the director’s arm length (*L*_di_), and the angle of the director’s arm (*AG*_di_). The optimal *D*_di_, *L*_di_, and *AG*_di_ were 25 mm, 43.34 mm, and 34.6 mm, respectively, covering the target operating frequency band of 470–890 MHz.

Figure 15a,b shows the simulated impedance bandwidth (|*S*_11_| ≤ −10 dB) under variable balun height (*H*_balun_) and distance from the center to the center of the balun (*D*_balun_). The optimal *H*_balun_ and *D*_balun_ were 140.32 mm and 19.24 mm, respectively. Meanwhile, the antenna height (*h*_di_; 156 mm) and the spacing between the lower base plate and the reflector (*h*; 139 mm) were proportional to the balun height (*H*_balun_) (Table 1).

## 4. Prototype Fabrication and Measured Results

Figure 16a,b depicts the prototype of the portable wideband directional antenna scheme for digital terrestrial television reception without and with ABS radome. Experiments with the antenna prototype (75 Ω input impedance) were carried out in an anechoic chamber using a 50 Ω vector network analyzer (Agilent E5061B) [34]. The impedance transformer (TME ZT-205) was used to match a 75 Ω antenna with a 50 Ω measurement system.

Figure 17 compares the simulated and measured impedance bandwidth (|*S*_11_| ≤ −10 dB) of the portable wideband directional antenna scheme. The measured impedance bandwidth was 75.93%, covering the 424–943 MHz frequency band, with the measured antenna gain of 2.69–4.84 dBi (Figure 18).

As shown in Figure 19, the radiation pattern of the proposed portable wideband horizontally polarized directional antenna scheme was of unidirectionality. The measured xz- and yz-plane cross-polarizations were below −20 dB and −40 dB, respectively, with the corresponding HPBW of 150°, 159°, and 160° and 102°, 78°, and 102° at 470 MHz, 680 MHz, and 890 MHz, respectively. The discrepancy between the simulated and measured cross-polarization could be attributed to the use of self-trapping screws in the antenna prototype fabrication, as shown in Figure 16a. The measured xz- and yz-plane back lobe levels were below −9 dB. The experiments also revealed that the ABS radome had negligible impacts on the impedance bandwidth and the antenna gain, vis-à-vis the antenna scheme without the ABS radome. Table 2 tabulates the simulated and measured performance of the proposed portable wideband directional antenna scheme. Table 3 shows a comparison between previous works and the proposed portable wideband directional antenna scheme.

## 5. Indoor and Outdoor Application Testing

The proposed portable wideband directional antenna scheme was tested for the indoor and outdoor reception of digital terrestrial television signals in the 470–862 MHz frequency range. This research was conducted in Thailand, which adopted the DVB-T2 standard for digital terrestrial television broadcasting. As a result, the indoor and outdoor tests were carried out within the 470–862 MHz frequency band.

The transmitting station is located in Thailand’s capital Bangkok at the coordinates 13°45′16″ N 100°32′24″ E, with an effective isotropic radiated power (EIRP) of 50 kW. The broadcast range encompasses the capital and neighboring provinces in the central region, as indicated by the area in pink in Figure 20. The height of the transmission antenna was 328.4 m. The receiving antenna (i.e., antenna under test (AUT)) for both the indoor and outdoor testing was located at the coordinates 14°20′52″ N 100°33′55″ E, which is 65.73 km from the transmitting station. In the indoor and outdoor testing, the AUT was placed at a height of at least 10 m above ground. The transmitting and receiving (AUT) antennas were in the line of sight.

Figure 21 shows the experimental setup for the outdoor and indoor testing of the proposed portable wideband directional antenna scheme with a digital video broadcasting (DVB) signal receiver (Promax Ranger Neo^+^). The measurements were taken in a concrete building. Figure 22 depicts the outdoor reception performance of the proposed antenna scheme at the first multiplexer (MUX) of 514 MHz in Thailand for the DVB-T2 system [35]. The power obtained was 38.4 dBµV (−70.4 dBm), with a carrier to noise ratio (C/N) of 11.6 dB. The signal spectrum was of a Rician fading channel, rendering the proposed antenna scheme suitable for outdoor digital terrestrial television reception [34].

Figure 23 demonstrates the indoor reception performance of the proposed antenna scheme at 514 MHz. The power obtained was 26.6 dBµV (−82.2 dBm) with a C/N of 10.9 dB. The signal spectrum was of a Rayleigh fading channel, rendering the proposed antenna scheme suitable for indoor digital terrestrial television reception [36].

Figure 24 shows the experimental setup used to assess the reception performance of the proposed antenna scheme in actual use, whereby the AUT was connected by a 75 Ω RG6 coaxial cable to the set top box and a television set. The results indicate that the proposed antenna scheme is capable of receiving the transmitted signals from all television channels, as evidenced by high-quality images and sound.

## 6. Conclusions

This research proposed a portable wideband of a horizontally polarized directional antenna scheme with a radome for digital terrestrial television reception (470–890 MHz). The proposed antenna scheme was adapted from the Yagi-Uda antenna, consisting of a folded bowtie radiator, semicircular corrugated reflector, and V-shaped director. The radome was of thin cylindrical ABS material. Simulations were carried out to optimize the antenna parameters. An antenna prototype was fabricated, and experiments were undertaken to determine the antenna performance. The performance metrics included the impedance bandwidth (|*S*_11_| ≤ −10 dB), gain, and unidirectionality. The results indicated that the folded bowtie radiator enhanced the impedance bandwidth of the antenna. The corrugated reflector improved the impedance bandwidth and unidirectionality, and the V-shaped director increased the antenna gain. The measured impedance bandwidth of the proposed antenna scheme was 75.93%, covering 424–943 MHz, with the measured antenna gain of 2.69–4.84 dBi. The radiation pattern was unidirectional for the entire target operating frequency band. The measured xz- and yz-plane HPBWs were 150°, 159°, and 160° and 102°, 78°, and 102° at 470, 680, and 890 MHz, with the corresponding cross-polarization below −20 dB and −40 dB. In addition, the ABS radome had a negligible effect on the impedance bandwidth and the antenna gain, vis-à-vis the antenna scheme without the ABS radome. The outdoor and indoor testing was also carried out to assess the outdoor and indoor reception performance of the proposed antenna scheme. The power obtained for the outdoor test, at 514 MHz, was 38.4 dBµV (−70.4 dBm) with a C/N of 11.6 dB. Meanwhile, the power obtained for the indoor test was 26.6 dBµV (−82.2 dBm) with a C/N of 10.9 dB. Essentially, the proposed portable wideband directional antenna scheme is operationally suitable for indoor and outdoor digital terrestrial television reception. Moreover, the portable antenna with a radome serves as a home decorative accessory given its modern and stylish design.

## Figures and Tables

**Figure 1 sensors-22-05338-f001:**
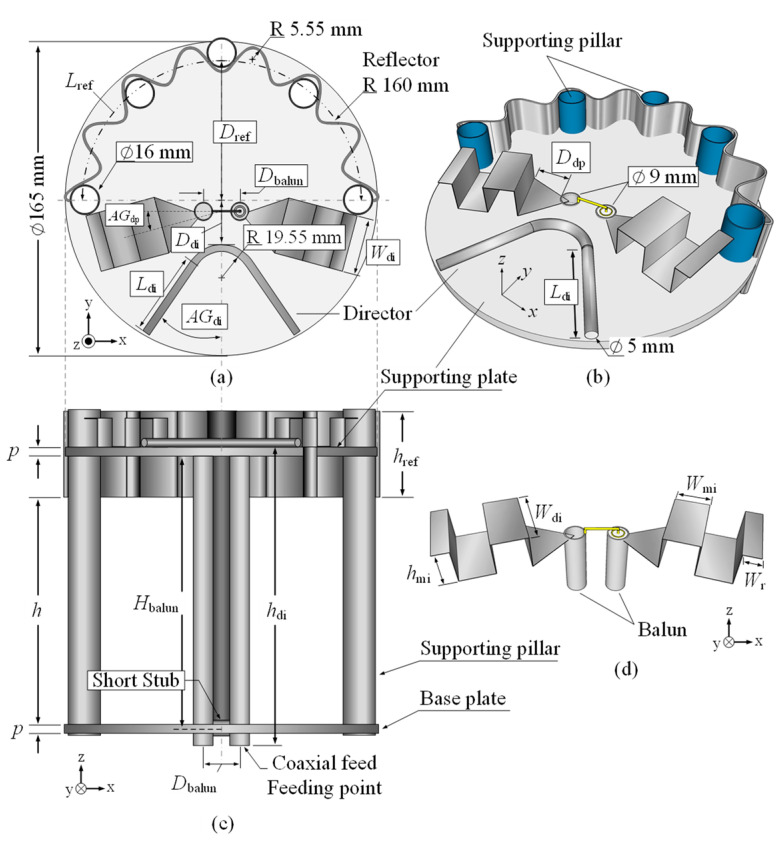
Geometry of the portable wideband directional antenna: (**a**) top view, (**b**) perspective view, (**c**) side view, and (**d**) folded bowtie radiator.

**Figure 2 sensors-22-05338-f002:**
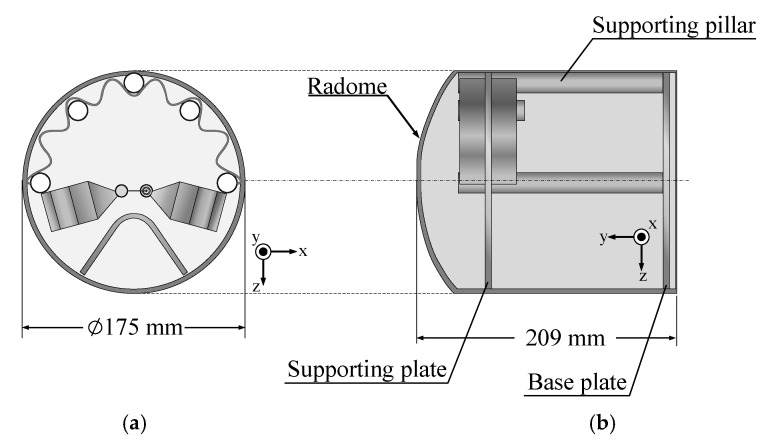
Geometry of the portable wideband directional antenna with radome: (**a**) top view, (**b**) side view.

**Figure 3 sensors-22-05338-f003:**
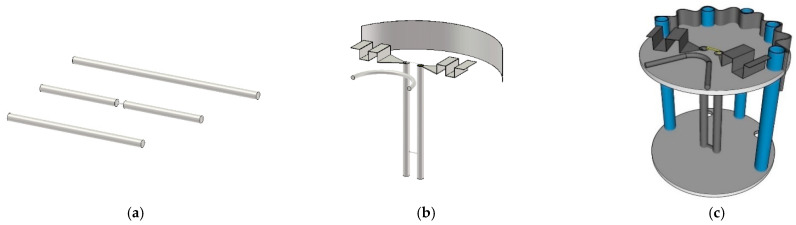
Conceptualization of the portable directional antenna scheme: (**a**) the Yagi-Uda antenna (model I), (**b**) the modified antenna with the U-shaped reflector and director (model II), and (**c**) the proposed portable antenna scheme (model III).

**Figure 4 sensors-22-05338-f004:**
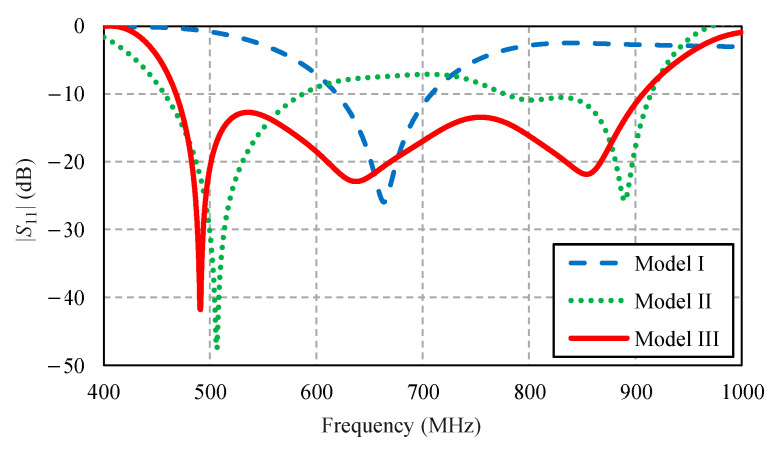
Simulated impedance bandwidth (|*S*_11_| ≤ −10 dB) of the antenna models I, II, and III.

**Figure 5 sensors-22-05338-f005:**
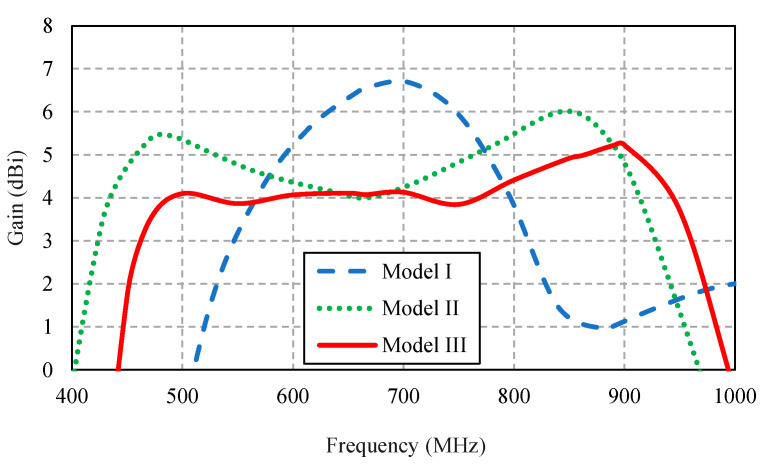
The simulated antenna gains of models I, II, and III.

**Figure 6 sensors-22-05338-f006:**
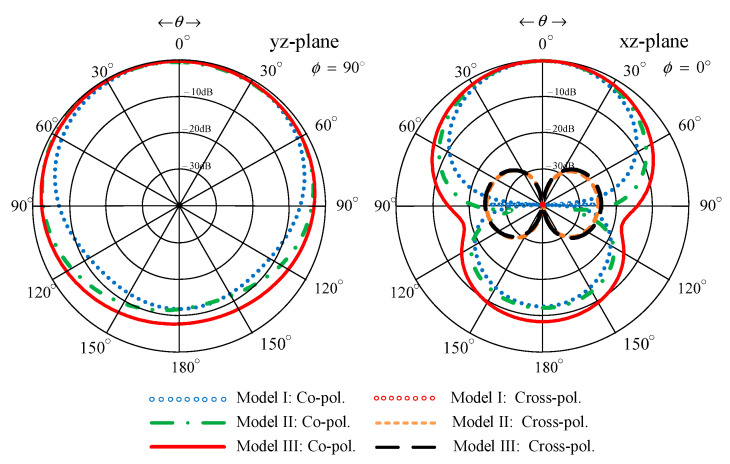
Simulated xz- and yz-plane radiation patterns of the antenna models I, II, and III.

**Figure 7 sensors-22-05338-f007:**
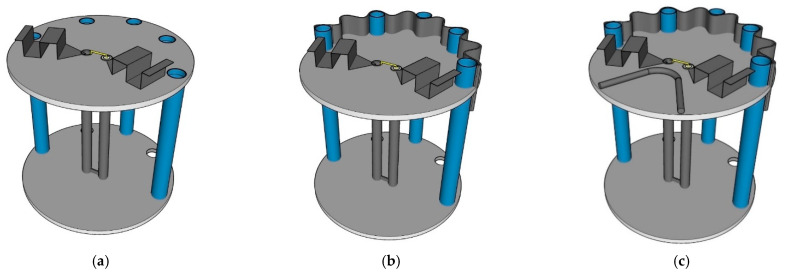
Evolutionary stages of the portable antenna scheme: (**a**) first generation, (**b**) second generation, and (**c**) third generation.

**Figure 8 sensors-22-05338-f008:**
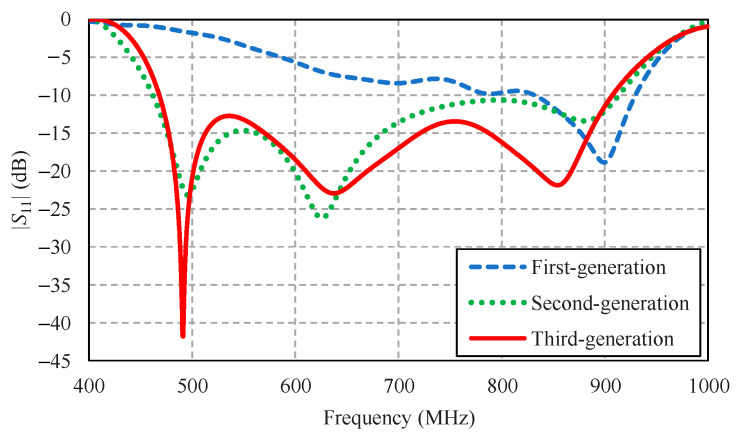
Simulated impedance bandwidth (|*S*_11_| ≤ −10 dB) of the first-, second-, and third-generation antenna schemes.

**Figure 9 sensors-22-05338-f009:**
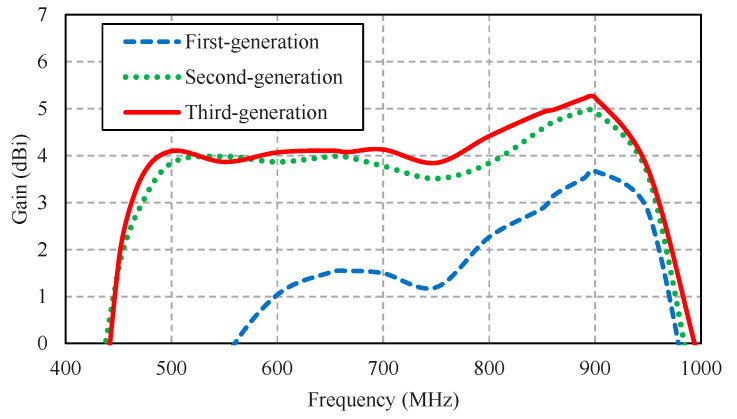
Simulated antenna gains of the first-, second-, and third-generation antenna schemes.

**Figure 10 sensors-22-05338-f010:**
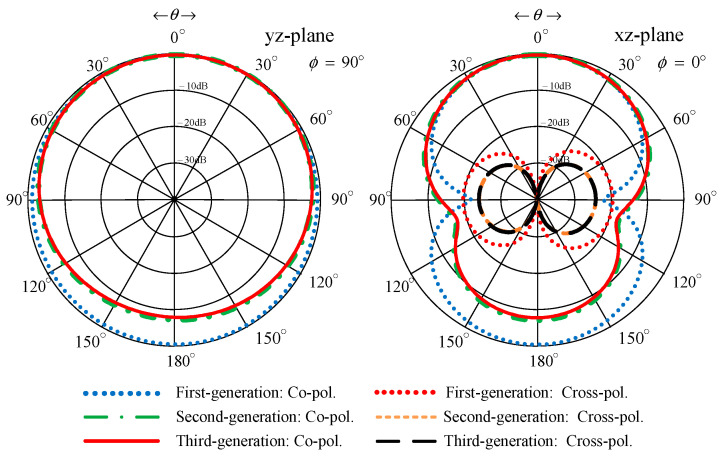
Simulated xz- and yz-plane radiation patterns of the first-, second-, and third-generation antenna schemes.

**Figure 11 sensors-22-05338-f011:**
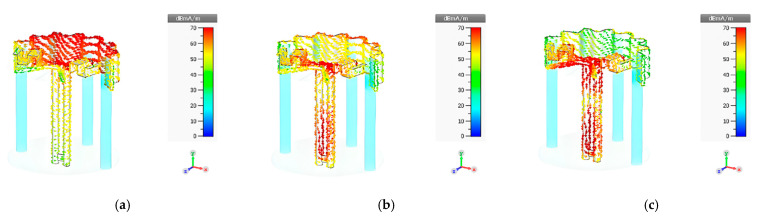
Simulated surface current distribution of the proposed portable wideband directional antenna scheme at: (**a**) 470 MHz, (**b**) 680 MHz, and (**c**) 890 MHz.

**Figure 12 sensors-22-05338-f012:**
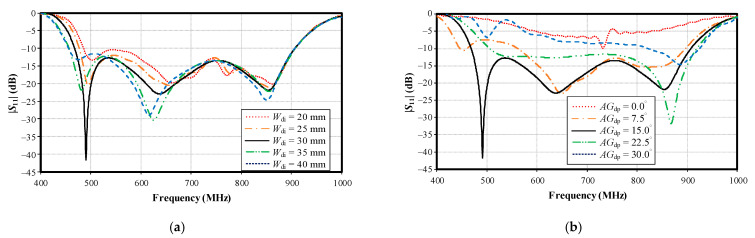
Simulated impedance bandwidth (|*S*_11_| ≤ −10 dB) under variable parameters: (**a**) width of folded bowtie radiator (*W*_di_), (**b**) angle of the folded bowtie radiator (*AG*_dp_).

**Figure 13 sensors-22-05338-f013:**
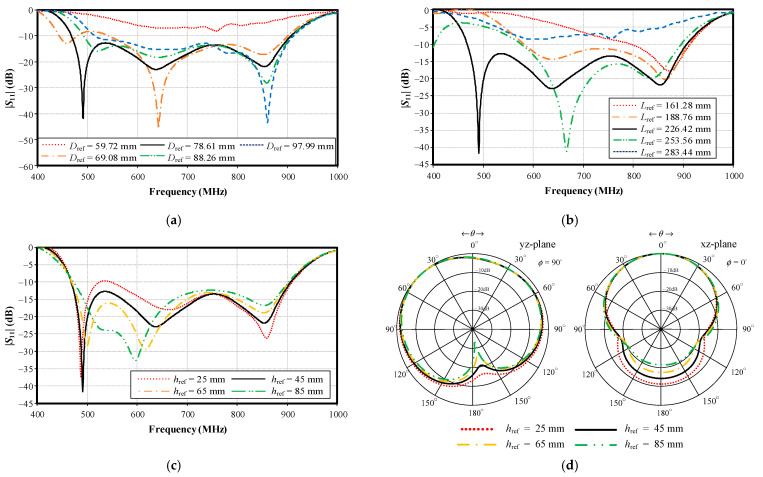
Simulated impedance bandwidth under variable parameters: (**a**) distance from center of the supporting plate to the reflector (*D*_ref_), (**b**) reflector length (*L*_ref_), (**c**) reflector height (*h*_ref_), and (**d**) radiation pattern at 890 MHz under variable *h*_ref_.

**Figure 14 sensors-22-05338-f014:**
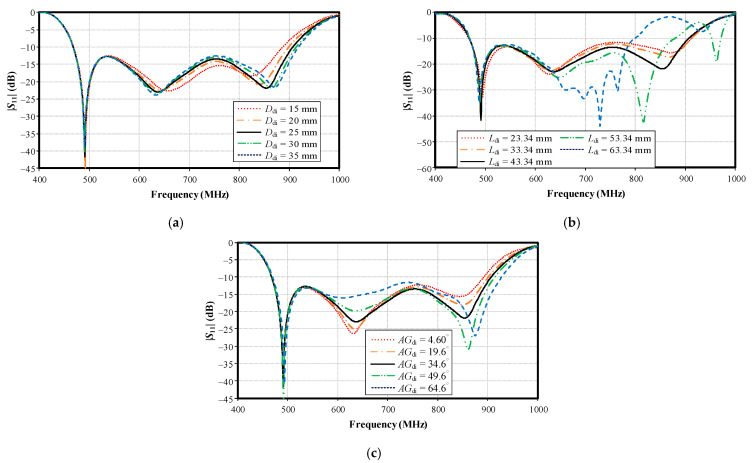
Simulated impedance bandwidth (|*S*_11_| ≤ −10 dB) under variable parameters: (**a**) distance between the director and center of the supporting plate (*D*_di_), (**b**) director’s arm length (*L*_di_), and (**c**) angle of the director’s arm (*AG*_di_).

**Figure 15 sensors-22-05338-f015:**
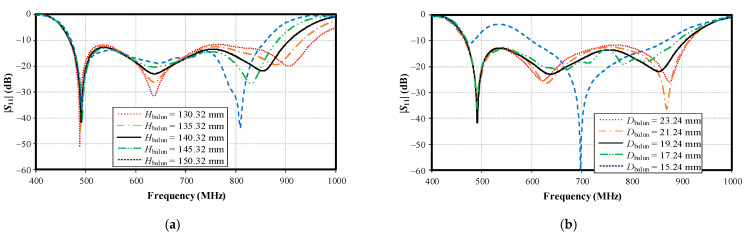
Simulated impedance bandwidth (|*S*_11_| ≤ −10 dB) under variable parameters: (**a**) balun height (*H*_balun_), (**b**) distance from center to center of the balun (*D*_balun_).

**Figure 16 sensors-22-05338-f016:**
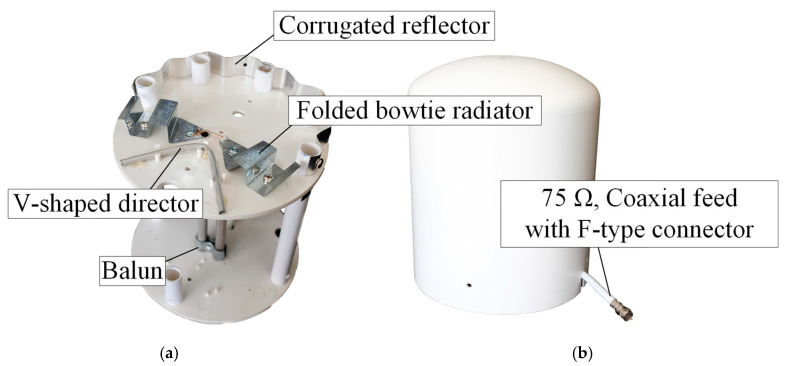
Prototype of the portable wideband directional antenna scheme: (**a**) without radome, (**b**) with radome.

**Figure 17 sensors-22-05338-f017:**
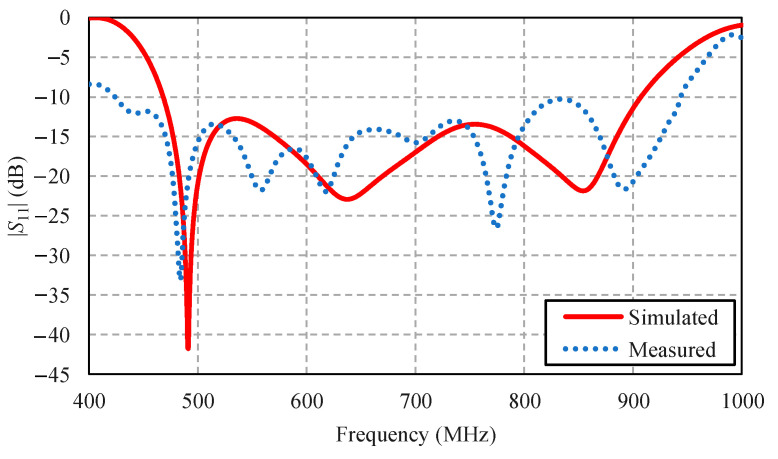
Simulated and measured impedance bandwidths of the portable wideband directional antenna scheme.

**Figure 18 sensors-22-05338-f018:**
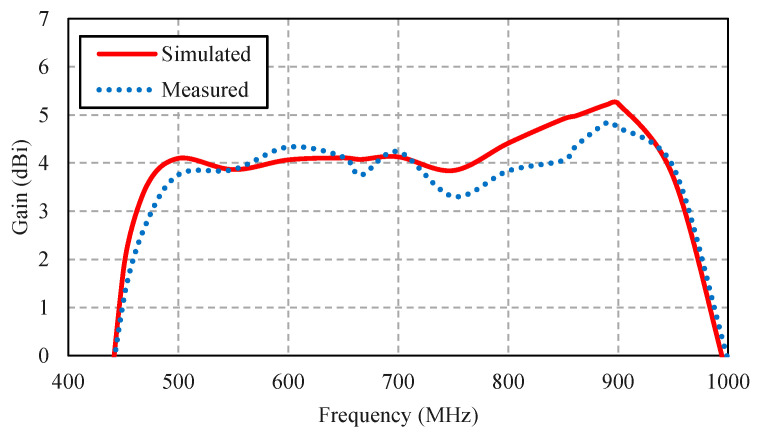
Simulated and measured gains of the portable wideband directional antenna scheme.

**Figure 19 sensors-22-05338-f019:**
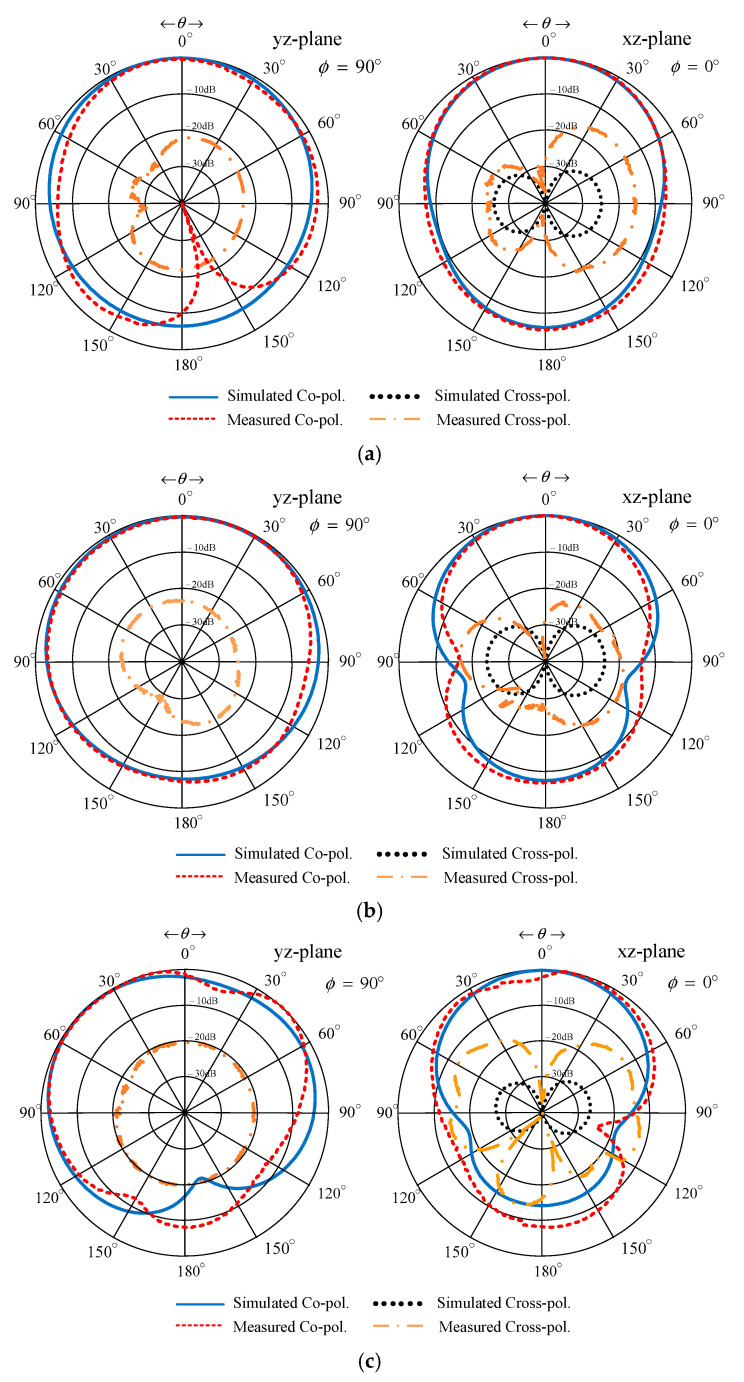
Simulated and measured xz- and yz-plane radiation patterns of the portable wideband directional antenna scheme: (**a**) 470 MHz, (**b**) 680 MHz, and (**c**) 890 MHz.

**Figure 20 sensors-22-05338-f020:**
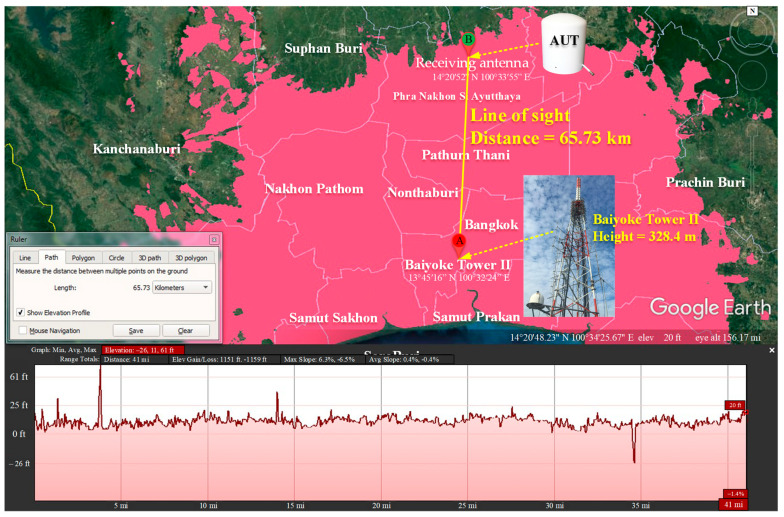
Distance between the transmitting and receiving antennas without obstructions (line of sight).

**Figure 21 sensors-22-05338-f021:**
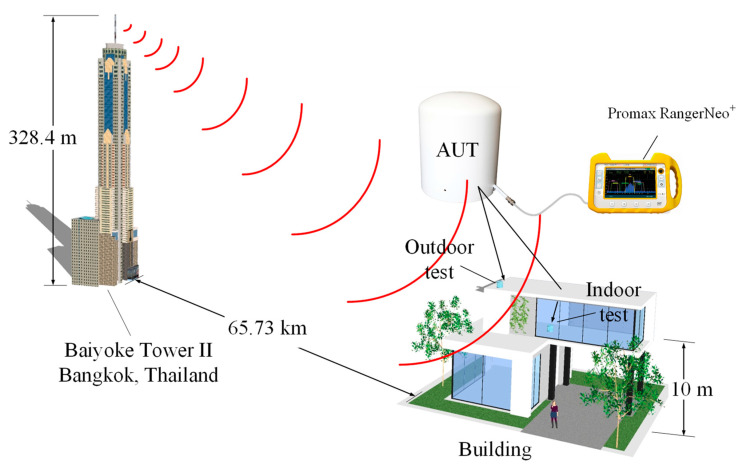
The portable wideband directional antenna scheme with a DVB signal receiver.

**Figure 22 sensors-22-05338-f022:**
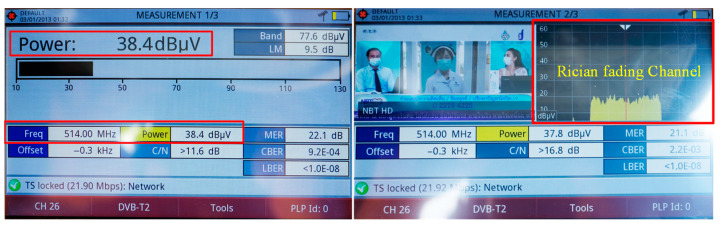
The outdoor reception performance of the portable wideband directional antenna scheme.

**Figure 23 sensors-22-05338-f023:**
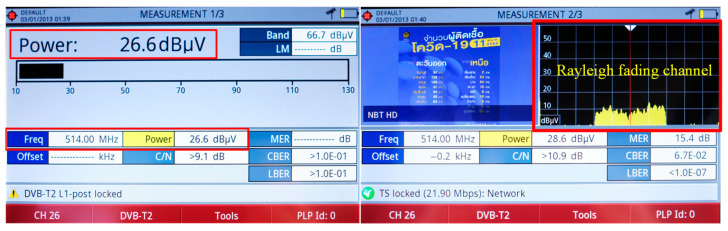
The indoor reception performance of the portable wideband directional antenna scheme.

**Figure 24 sensors-22-05338-f024:**
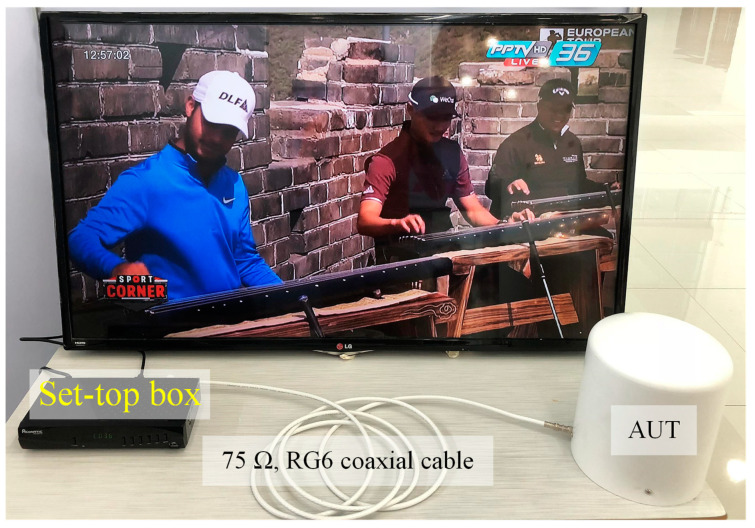
The experimental setup for the reception test in actual use.

**Table 1 sensors-22-05338-t001:** The optimal dimensions of the portable wideband directional antenna for digital terrestrial television reception.

Symbol	Description	Physical Size, mm	Electrical Size at 680 MHz
*h*	Spacing between the lower base plate and the reflector	139.00	0.315 λ_C_
*p*	Thickness of the supporting plate and the base plate	5.00	0.011 λ_C_
*H* _balun_	Balun height	140.32	0.318 λ_C_
*h* _di_	Antenna height	156.00	0.353 λ_C_
*D* _balun_	Distance between center to center of the balun	19.24	0.043 λ_C_
*h* _ref_	Reflector height	45.00	0.102 λ_C_
*L* _ref_	Reflector length	226.42	0.513 λ_C_
*D* _ref_	Distance from center of the supporting plate to the reflector	78.61	0.178 λ_C_
*AG* _dp_	Angle of the folded bowtie radiator (degree)	15.00	-
*W* _di_	Width of the folded bowtie radiator	30.00	0.068 λ_C_
*D* _di_	Distance between director and center of the supporting plate	25.00	0.056 λ_C_
*L* _di_	Director’s arm length	43.34	0.098 λ_C_
*AG* _di_	Angle of the director’s arm (degree)	34.60	-
*D* _dp_	Length of triangular section of the folded bowtie radiator	18.70	0.042 λ_C_
*h* _mi_	Height of the folded bowtie radiator	15.00	0.034 λ_C_
*W* _mi_	Width of the 1st and 2nd sections of the folded bowtie radiator	15.20	0.034 λ_C_
*W* _r_	Width of the final section of the folded bowtie radiator	6.90	0.015 λ_C_

λ_C_ is the wavelength at the center frequency of the target operation band (470–890 MHz).

**Table 2 sensors-22-05338-t002:** Simulated and measured performances of the portable wideband directional antenna scheme.

Specifics	Simulation			Measurement		
	470 MHz	680 MHz	890 MHz	470 MHz	680 MHz	890 MHz
|*S*_11_| < −10 dB Bandwidth, %	(468–907), 63.85%			(424–943), 75.93%		
Gain (dBi)	3.45	4.07	5.21	2.69	3.75	4.84
HPBW in xz-plane, (deg.)	149	194	185.1	150	159	160
HPBW in yz-plane, (deg.)	99	92.8	84.2	102	78	102
Cross-pol. in xz-plane (dB)	≤−40			≤−20		
Cross-pol. in yz-plane (dB)	≤−40			≤−40		
Back lobe level (dB)	≤−10			≤−9		
Radiation pattern	Unidirectional			Unidirectional		

**Table 3 sensors-22-05338-t003:** Comparison between previous works and the proposed portable wideband directional antenna scheme.

Reference	Antenna Type	Bandwidth (MHz), (%)	Gain (dBi)	Dimension (mm) (Electrical Size at 470 MHz)
[1]	Printed dipole with multiple loops antenna	430–1180 (93.16%)	1.78–2.83	58.5 × 241 × 1.6 (0.091λ × 0.377λ × 0.0025λ)
[2]	Printed monopole with spiral loops antenna	470–862 (58.85%)	−19–(−12)	20 × 30 × 0.4 (0.031λ × 0.047λ × 0.0006λ)
[3]	PIFA-loop antenna	470–771 (48.50%)	2.2–4.8	17 × 375 (0.026λ × 0.587λ)
[4]	Printed meander line antenna with magneto dielectric	467–1012 (73.69%)	0.86–1.65	10 × 25 × 1 (0.015λ × 0.039λ × 0.0015λ)
[19]	Printed dipole antenna	455–1070 (80.65%)	−0.57–1.15	20 × 200 × 0.8 (0.031λ × 0.313λ × 0.001λ)
[20]	Printed dipole antenna	452–897 (65.97%)	2.09–3.85	45 × 250 × 1.6 (0.070λ × 0.391λ × 0.002λ)
[21]	Printed dipole antenna	441–890 (67.46%)	4.65 (peak)	35 × 243 × 1.6 (0.054λ × 0.380λ × 0.002λ)
[23]	Printed loop antenna	461–806 (54.45%)	1.9–2.5	165 × 165 × 0.8 (0.258λ × 0.258λ × 0.001λ)
[24]	Printed quasi-Yagi antenna	450–848 (61.32%)	3.5–4.6	200 × 240 × 1.6 (0.313λ × 0.376λ× 0.002λ)
[26]	Log-periodic antenna	470–790 (50.79%)	8 (peak)	302.6 × 356 × 35 (0.474λ × 0.557λ 0.054λ)
Proposed antenna	Yagi-Uda with bowtie antenna	424–943 (75.93%)	2.69–4.84	175 × 175 × 209 (0.274λ × 0.274λ × 0.327λ)

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
