# Peer review of "Portable Wideband Directional Antenna Scheme with Semicircular Corrugated Reflector for Digital Television Reception"

_sensors, 2022, doi:10.3390/s22145338_

Round 1
Reviewer 1 Report
Thank you for your detailed work on Wideband Directional Antenna Scheme having Corrugated Reflector.
The introduction is well explained and up to date.
The geometry of the antenna is well explained by proper dimensions. The reader will find it interesting.
A parametric study is performed for all antenna parameters including reflection coefficient, radiation pattern, and antenna gain. (this is very rare in antenna papers and usually, authors do it for reflection coefficient, which is not a good way to handle antenna).
All figures are made of high quality and are self-explanatory.
Figure 11: the color bar needs to be updated for the other two plots as well. As I can see only one color bar.
Simulated and measured results agree.
This is a high-quality antenna paper and needs to be accepted in the current format.
Thanks for such a fine contribution.
Reviewer 2 Report
1. Please add a comparison table with a figure of merits for the recently reported antennas for this application.
2. Can the authors clarify if the corrugated reflector with this configuration is newly proposed?. Please add the original reference of the corrugated reflector.
3. Fig. 19 reveals that the measured cross-pol. is much worse than the simulated one. Can the authors comment on this phenomenon?
Reviewer 3 Report
The submitted manuscript proposes a wideband directional antenna employing a corrugated reflector. The design and the results are interesting and the manuscript is well organized, and clear. The front-to-back ratio is still small, please show the simulated results of changing the height of the reflector "h_ref" as it may have some effect on that. The authors are also invited to think about other methods to improve the front-to-back ratio if increasing h_ref does not work well.
Round 2
Reviewer 3 Report
The revised manuscript is acceptable now. One minor change, please revise the value of h_ref in Fig. 13(d).
